# Medical relevance of protein-truncating variants across 337,205 individuals in the UK Biobank study

Christopher DeBoever[1,2], Yosuke Tanigawa [1], Malene E. Lindholm[3], Greg McInnes[1], Adam Lavertu[1], Erik Ingelsson[4], Chris Chang[3], Euan A. Ashley [5], Carlos D. Bustamante[1,2], Mark J. Daly [6,7] & Manuel A. Rivas[1]

Protein-truncating variants can have profound effects on gene function and are critical for clinical genome interpretation and generating therapeutic hypotheses, but their relevance to medical phenotypes has not been systematically assessed. Here, we characterize the effect of 18,228 protein-truncating variants across 135 phenotypes from the UK Biobank and find 27 associations between medical phenotypes and protein-truncating variants in genes outside the major histocompatibility complex. We perform phenome-wide analyses and directly measure the effect in homozygous carriers, commonly referred to as "human knockouts," across medical phenotypes for genes implicated as being protective against disease or associated with at least one phenotype in our study. We find several genes with strong pleiotropic or non-additive effects. Our results illustrate the importance of protein-truncating variants in a variety of diseases.

[1] Department of Biomedical Data Science, Stanford University, Stanford, CA 94305, USA. [2] Department of Genetics, Stanford University, Stanford, CA 94305, USA. [3] Grail, Inc., 1525 O'Brien Drive, Menlo Park, CA 94025, USA. [4] Division of Cardiovascular Medicine, Department of Medicine, Stanford University School of Medicine, Stanford, CA 94305, USA. [5] Department of Medicine, School of Medicine, Stanford University, Stanford, CA 94305, USA. [6] Analytical and Translational Genetics Unit, Boston, MA 02114, USA. [7] Broad Institute of MIT and Harvard, Cambridge 02142 MA, USA. Correspondence and requests for materials should be addressed to M.A.R. (email: mrivas@stanford.edu)

Protein-truncating variants (PTVs), genetic variants predicted to shorten the coding sequence of genes, are a promising set of variants for drug discovery since identification of PTVs that protect against human disease provides in vivo validation of therapeutic targets[1–4]. Although tens of thousands of germline PTVs have been identified[5–9], their medical relevance across a broad range of phenotypes has not been characterized. Because most PTVs are present at low frequency, assessing the effects of PTVs requires genotype data from many individuals with linked phenotype data for a variety of diseases and physiological measurements. The recent release of genotype and linked clinical and questionnaire data for 488,377 individuals in the UK Biobank provides an unprecedented opportunity to assess the clinical impact of truncating protein-coding genes at a resolution not previously possible.

PTVs are genetic variants that disrupt transcription and lead to a shortened or absent protein that often causes loss of protein function though it is also possible to observe gain-of-function effects[10]. PTVs include nonsense single-nucleotide variants (SNVs), frameshift insertions or deletions (indels), large structural variants, and splice-disrupting SNVs[8]. Although most common genetic variants associated with disease have relatively small effect on disease risk, PTVs are expected to have much stronger effects on disease risk as they dramatically alter protein sequence[11]. Population sequencing efforts have estimated that every human genome contains ~ 100 PTVs although this rate can be higher in consanguineous populations[6,12,13]. Prior studies have identified a number of associations between PTVs and disease risk. PTVs that are associated with protection against disease are particularly interesting as they indicate genes that may be targeted for therapeutics. For instance, PTVs in *CARD9*, *RNF186* and *IL23R* provide protection against Crohn's disease and/or ulcerative colitis[1,2] and PTVs in *ANGPTL4*, *PCSK9*, *LPA*, and *APOC3* protect against coronary heart disease[4,13–17].

Here, we test for associations between PTVs and 135 different medical phenotypes including cancers and complex diseases among 337,205 participants in the UK Biobank. We identify 27 PTVs outside of the MHC that are associated with at least one medical phenotype, including several protective associations. We perform phenome-wide association analyses across 206 medical phenotypes for these PTVs as well as PTVs with previously identified associations and find PTVs with pleiotropic effects. We also perform a human "knockout" analysis to identify non-additive associations for homozygous or compound heterozygous PTV carriers and find several genes with non-additive effects. The associations reported here indicate new disease-causing genes that may be promising therapeutic targets.

## Results

**PTV genetic association analysis.** To assess the clinical relevance of PTVs, we cataloged predicted PTVs present on the Affymetrix UK Biobank array and their effects on medical phenotypes from 337,205 unrelated individuals in the UK Biobank study[18,19]. We defined PTVs as SNVs predicted to introduce a premature stop codon or to disrupt a splice site or small indels predicted to disrupt a transcript's reading frame. Although methods to predict PTVs, also referred to as loss-of-function or knockout variants[6,20], are still being improved and validated[21], previous work has found that 70% of nonsense PTVs predicted to cause nonsense-mediated decay show evidence for decreased expression of the corresponding transcript and 79% of splice-site variants disrupt splicing[8], indicating that predicted PTVs are likely to affect gene expression or function.

We identified 18,228 predicted PTVs in the UK Biobank array that were polymorphic across 8750 genes after filtering (Methods, Supplementary Fig. 1). Each participant had 95 predicted PTVs with minor allele frequency (MAF) < 1% on average, and 778 genes were predicted to be homozygous or compound heterozygous for PTVs with MAF < 1% in at least one individual. We observed 291 genes that had at least one observed homozygous PTV carrier in our study but had no observed homozygous loss-of-function carriers in previous studies (Supplementary Data 1). The observed number of PTVs per individual is consistent with the ~ 100 loss-of-function variants observed in the 1000 Genomes project[22]. In contrast, the number of PTV singletons (or observed allele counts < 10) in ExAC suggests approximately five singletons per individual and only ~ 0.2 per individual in highly constrained genes[9,23]. These observations indicate that the majority of PTVs in an individual are common (or common and low frequency) such that they can be assessed via genotyping.

We used computational matching and manual curation based on hospital in-patient data (National Health Service Hospital Episode Statistics), self-reported verbal questionnaire data, and cancer and death registry data to define a broad set of medical phenotypes including various cancers, cardiometabolic diseases, and autoimmune diseases (Supplementary Data 2)[24]. We then performed association analyses between the 3724 PTVs with MAF > 0.01% and 135 medical phenotypes with at least 2000 case samples (Fig. 1, Supplementary Fig. 2) and stratified the association results into three bins based on PTV MAF > 1% (463 PTVs), between 0.1% and 1% (700 PTVs), and between 0.01% and 0.1% (2561 PTVs) to account for expected differences in the statistical power to detect associations for PTVs with different MAFs (Supplementary Fig. 3). We adjusted the nominal association *p* values separately for each MAF bin using the

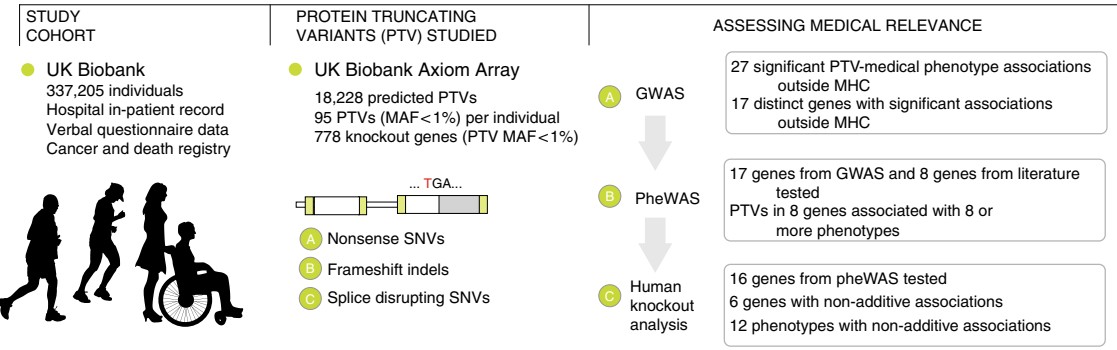

**Fig. 1** Schematic overview of the study. We prepared a data set of 18,228 protein-truncating variants and 135 medical phenotypes from the UK Biobank data set of 337,205 individuals. From these data, we analyzed the clinical effects of predicted protein-truncating genetic variants

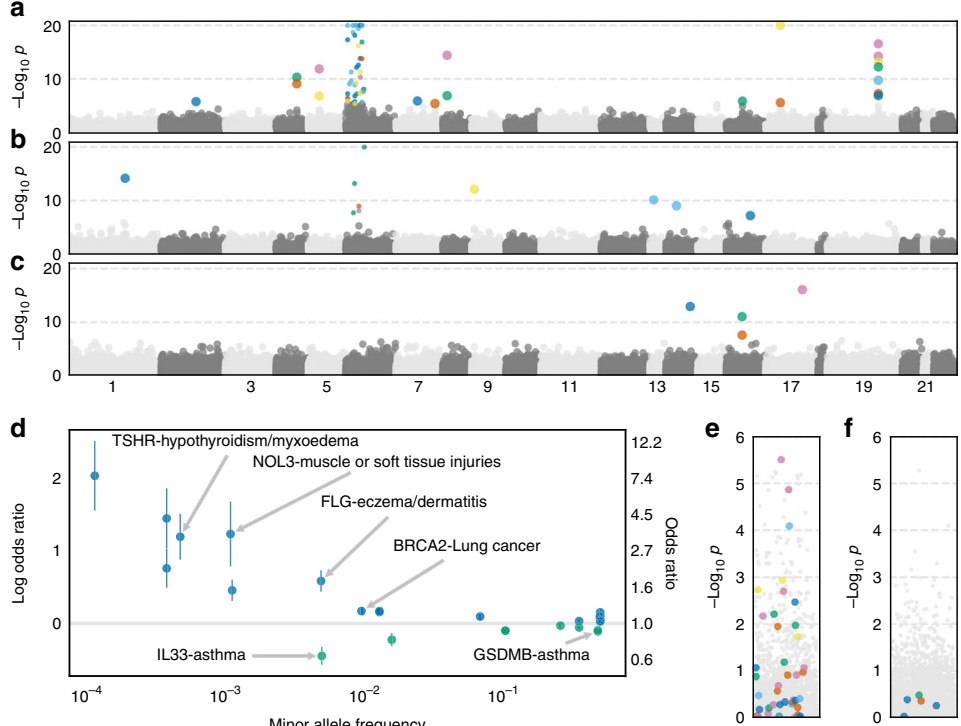

**Fig. 2** Identification of risk and protective alleles for 135 phenotypes. **a–c** Manhattan plots for logistic regression for all PTVs and all phenotypes stratified by minor allele frequency **a** > 1%, **b** between 0.1% and 1%, and **c** between 0.01% and 0.1%. Scatter points are colored according to phenotype. Fourteen associations with -log$_{10}$ p values> 20 were plotted at 20. PTVs in genes near or in the MHC region have smaller scatter points. **d** Effect size "cascade plot" for all associations outside the MHC with BY-adjusted p < 0.05. Error bars represent 95% confidence intervals. **e–f** Manhattan plots for PTVs in or near the MHC with minor allele frequency **e** > 1% and **f** between 0.1 and 1%. The p values for gray points are the same as in **a** and **b**, respectively. The p values for the color points have been re-calculated conditional on HLA alleles

Benjamini-Yekutieli (BY) procedure to correct for multiple hypothesis testing and identified 74 significant associations between PTVs and medical phenotypes (BY-adjusted p < 0.05, Fig. 2a–c, Supplementary Data 3).

Among the 74 PTV-phenotype associations we identified, 27 involved PTVs in genes outside of the MHC (chr6–25, 477, 797–36, 448, 354). As PTVs in or near the MHC likely tag HLA risk alleles, we focused on associations for PTVs outside of the MHC. We identified five PTVs with seven associations consistent with protective effects (odds ratio (OR) < 1, BY-adjusted p < 0.05, Fig. 2d, Supplementary Data 3). We found that the rare splice-disrupting PTV rs146597587 in *IL33* is associated with protection against asthma (MAF = 0.48%, $p = 7.6 \times 10^{-13}$, OR = 0.64, 95% confidence interval (CI): 0.57–0.72). This PTV is negatively associated with eosinophil counts ($\beta = -0.21$ SD, $p = 2.5 \times 10^{-16}$) and has suggestive evidence of an association with asthma ($p = 1.8 \times 10^{-4}$, OR = 0.47, 95% CI: 0.32–0.70)[25]. Our results provide strong evidence in an independent sample that this PTV protects against asthma and suggests that knocking down *IL33* function may be a useful therapeutic approach for asthma. We also identified protective associations for the PTV rs11078928 (MAF = 47.1%) in *GSDMB* against asthma ($p = 6.3 \times 10^{-50}$, OR = 0.90, 95% CI: 0.88–0.91) and bronchitis ($p = 2.6 \times 10^{-6}$, OR = 0.91, 95% CI: 0.87–0.95). *GSDMB* is associated with asthma in humans and induces an asthma phenotype in mouse when over-expressed[26,27]. We identified additional protective associations between PTVs in *IFIH1* and hypothyroidism (labeled as hypothyroidism/myxedema) (MAF = 1.5%, $p = 1.7 \times 10^{-6}$, OR = 0.80, 95% CI: 0.73–0.88) and *VKORC1* and hypertension (MAF = 25.3%, $p = 1.4 \times 10^{-6}$, OR = 0.97, 95% CI: 0.96–0.98).

We also found 20 risk associations for PTVs in 12 genes outside the MHC (Fig. 2d, Supplementary Data 3). We identified clinically relevant PTV-phenotype associations such as *FLG*, whose protein product contributes to the structure of epidermal cells, and eczema/dermatitis (MAF = 0.48%, $p = 6.7 \times 10^{-15}$, OR = 1.80, 95% CI: 1.55–2.08)[28] and *TSHR*, thyroid-stimulating hormone receptor, and hypothyroidism/myxedema (MAF = 0.046%, $p = 1.2 \times 10^{-13}$, OR = 3.30, 95% CI: 2.41–4.53)[29]. We replicated known risk genome-wide association study (GWAS) associations such as *BRCA2* and family history of lung cancer (MAF = 0.93%, $p = 7.3 \times 10^{-11}$, OR = 1.19, 95% CI: 1.13–1.25)[30] and rs33966350 in *ENPEP* and hypertension (MAF = 1.3%, $p = 4.8 \times 10^{-11}$, OR = 1.17, 95% CI: 1.12–1.23)[31] and identified risk associations between *FANCM*, a member of the same gene family as *BRCA2*, and lung cancer (MAF = 0.11%, $p = 9.7 \times 10^{-10}$, OR = 1.58, 95% CI: 1.36–1.83) as well as *NOL3*, a regulator of apoptosis in muscle cells, and muscle or soft tissue injury (MAF = 0.11%, $p = 6.5 \times 10^{-8}$, OR = 3.43, 95% CI: 2.19–5.36)[32,33]. To investigate the association between *NOL3* and tissue injury, we knocked down *NOL3* threefold in differentiated human skeletal muscle cells and used electrical pulse stimulation to induce cell damage and simulate injury. Lower expression of NOL3 resulted in increased activation of caspase 8, an early indicator of apoptosis, in the damaged cells, consistent with the observation that NOL3 inhibits caspase 8 (Supplementary Fig. 4a, b)[34]. The degree of DNA fragmentation, another indicator of tissue damage, was also higher in NOL3 knockdown cells compared with control (Supplementary Fig. 4c). We observed higher expression of MAFbx/atrogin-1 (mRNA and protein), a muscle-specific E3 ubiquitin ligase that is activated during skeletal

muscle atrophy[35], in *NOL3* knockdowns without stimulation (Supplementary Fig. 4d, e), consistent with increased expression of MAFbx in NOL3 knockout mice[36] and general protein degradation after stimulation. These results provide additional evidence that *NOL3* has an important role in muscle injury.

Even in the context of variants with strong predicted effects such as PTVs, it is critical to evaluate whether the associated variant is causal in the context of neighboring variants. We initially identified an association between the PTV rs34358 in *ANKDD1B* and high cholesterol, although this association disappeared upon conditional analysis with rs17238484, an intronic variant in *HMGCR* known to be associated with cholesterol levels[37]. Another association between rs34358 and family history of diabetes remained upon conditional analysis with rs17238484 ($p = 9.1 \times 10^{-5}$, OR = 1.03, 95% CI: 1.02–1.05). We performed conditional analyses for the remaining 27 associations outside of the MHC by identifying genotyped variants within 10 kb of the associated PTV and using the genotypes of the nearby variants as covariates for logistic regression. For PTVs with MAF < 1%, we found that only the association between a PTV in *HEATR6* and retinal detachment was explained by a nearby variant rs3744375 (Supplementary Data 3). Six of the common (MAF > 1%) PTVs with associations were in high linkage disequilibrium with other nearby common variants that may explain the observed associations (Supplementary Data 3, Supplementary Fig. 5), though the PTVs remain strong functional candidate for these associations. For instance, the gain-of-function PTV rs328 in *LPL* (MAF = 10.1%) that we find to be associated with decreased risk for high cholesterol ($p = 3.9 \times 10^{-15}$, OR = 0.90, 95% CI: 0.88–0.93) and angina ($p = 1.3 \times 10^{-7}$, OR = 0.91, 95% CI: 0.87–0.94) has been associated with coronary artery disease, lipid metabolism, and lower triglyceride levels[17,38,39]. Similarly, a recent study found that the PTV rs11078928 in *GSDMB* that offers protection against asthma removes exon 6 from the transcript and eliminates the ability of *GSDMB* to induce cell death[40]. The PTV rs2004640 in *IRF5* has previously been associated with rheumatoid arthritis and has been connected to pathogenesis in the mouse model[41,42]. The PTV rs601338 in *FUT2* determines secretor status for ABH blood groups that has been associated with susceptibility to infection and several diseases[43–47]. The PTV rs2884737 in *VKORC1* associated with hypertension is in moderate LD ($R^2 \approx 0.56$) with several nearby common variants and the PTV rs776746 in *CYP3A5* associated with hayfever/allergic rhinitis is in near perfect LD with one other nearby variant. Additional functional work may be needed to establish whether the PTVs are causal for these two associations.

We identified five significant associations between PTVs and family history phenotypes included in our analysis (Supplementary Data 3). For two of these associations, the variant associated with the family history phenotype was also associated directly with the phenotype. rs180177132 in *PALB2* was associated with a family history of breast cancer (MAF = 0.037%, $p = 2.5 \times 10^{-8}$; OR = 2.14, 95% CI: 1.64–2.79) as well as breast cancer diagnosis ($p = 9.0 \times 10^{-12}$; OR = 4.25, 95% CI: 2.80–6.43) and *FUT2* was associated with family history of high blood pressure (MAF = 49.1%, $p = 1.3 \times 10^{-7}$; OR = 1.03, 95% CI: 1.02–1.04), hypertension diagnosis ($p = 5.7 \times 10^{-13}$; OR = 1.04, 95% CI: 1.03–1.05), and essential hypertension ($p = 5.2 \times 10^{-8}$, OR = 1.04, 95% CI: 1.02–1.05). We also found that the PTV rs11571833 in *BRCA2* was associated with lung cancer (MAF = 0.934%, $p = 7.3 \times 10^{-11}$, OR = 1.19, 95% CI: 1.13–1.25). These results demonstrate previous approaches for identifying genetic associations using family history information (e.g., ref. [48,49]) can be applied even to relatively rare PTVs.

To further characterize the PTV-phenotype associations, we asked whether missense variants with MAF > 0.01% in the genes with significant PTV associations were also associated with the same phenotypes. For each of the 27 PTV-phenotype associations in our GWAS, we performed association analyses between the missense variants in that gene and the phenotype that the PTV was associated with and found 23 missense variant-phenotype associations with $p < 0.001$ (Supplementary Data 3). Thirteen of these 23 associations remain significant after a conditional analysis including the PTV genotype as a covariate indicating that several genes with PTV associations also contain independent missense associations. For instance, we found two different missense variants in *TSHR* that were both associated with hypothyroidism independent of the PTV association. We also identified independent missense associations for genes and phenotypes such as *ENPEP* and hypertension; *GSDMB* and asthma; *IFIH1* and hypothyroidism; and *PALB2* and lung cancer (Supplementary Data 3). In total, we found at least one missense association for seven genes implicated in our PTV GWAS providing more evidence that these genes are likely important to the etiology of these conditions.

Forty-seven of the 74 significant associations involved PTVs in genes in or near the MHC (Supplementary Data 4). 4). To investigate whether these associations are caused by linkage between these PTVs and HLA susceptibility alleles, we performed association analyses for each of these PTVs conditional on the presence of each of 344 HLA alleles that were polymorphic among the 337,205 subjects (Supplementary Data 4). (Supplementary Data 4). We found that the $p$ values for all five associations with MAF between 0.1 and 1% were > 0.05 for at least one HLA allele (Fig. 2e). Similarly, the $p$ values for 30 of 42 associations with MAF > 1% were > 0.05 for at least one HLA allele and only three were < 0.001 (Fig. 2f). For instance, we identified an association between rs72841509 in *BTN3A2* and Celiac disease (coded malabsorption/celiac disease) in our initial GWAS (MAF = 0.13, $p = 1.8 \times 10^{-119}$, OR = 2.33, 95% CI: 2.17–2.50). However, conditioning upon the presence of HLA-B8, which is on the same haplotype as the HLA-DQ2 Celiac risk allele, reduced the $p$ value of the association between rs72841509 and Celiac disease to $p = 0.92$[50,51]. These results indicate that the majority of the associations identified here for PTVs in MHC genes are likely due to LD with HLA susceptibility alleles and show that it is important to carefully consider the genomic context of associated variants, even for variants with strong predicted effects[52].

We next investigated whether we could identify PTV-phenotype associations using imputed genotypes. After filtering (Methods), we identified 546 PTVs outside the MHC with MAF greater than 0.01% among the UK Biobank imputed genotypes. We stratified these PTVs into the same MAF bins as above (0.01–0.1%, 0.1–1%, and 1–50%) and applied the BY adjustment to the association $p$ values for each bin. We found nine significant associations for imputed PTVs (BY-adjusted $p < 0.05$, Supplementary Data 3) including rs74315329 in *MYOC* and glaucoma (MAF = 0.0012, $p = 1.8 \times 10^{-30}$, OR = 4.71, 95% CI: 3.61–6.14)[53], a well-known risk variant for glaucoma[54], and *D2HGDH* and asthma (MAF = 0.445, $p = 1.6 \times 10^{-12}$, OR = 0.95, 95% CI: 0.94–0.96) and hayfever (coded hayfever/allergic rhinitis) ($p = 8.4 \times 10^{-9}$, OR = 0.94, 95% CI: 0.92–0.96). The *D2HGDH* PTV is in partial LD with an intronic variant rs34290285 in *D2HGDH* ($r^2 = 0.366$, LDlink) that has been associated with asthma and allergic disease in the initial UK Biobank data release[55,56]. We also identified an association between the PTV rs754512 in *MAPT* and Parkinson's disease (MAF = 0.23, $p = 1.1 \times 10^{-6}$; OR = 0.94, 95% CI: 0.92–0.97)[57]. This variant is predicted to be a PTV but is in the intron of the canonical *MAPT* transcript and

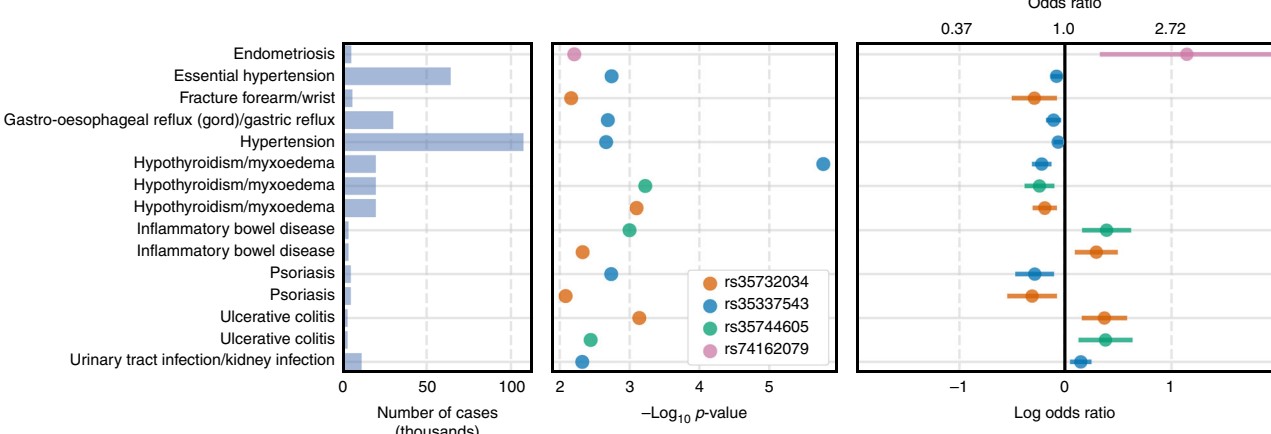

**Fig. 3** PheWAS for IFIH1. Phenome-wide associations (logistic regression, $p < 0.01$) for four PTVs in IFIH1 with minor allele frequency > 0.01%. The left panel shows the number of cases per phenotype in thousands. The middle panel shows the logistic regression $-\log_{10} p$ values. The right panel shows the estimated odds ratios and 95% confidence intervals

lies on the same haplotype as three *MAPT* missense variants (rs17651549, rs62063786, rs10445337) so conditional analysis could not establish the causal allele. We found associations between a PTV in *RPL3L* and atrial flutter (MAF = 0.0021, $p = 5.0 \times 10^{-10}$, OR = 0.54, 95% CI: 0.44–0.66) and atrial fibrillation ($p = 2.3 \times 10^{-9}$, OR = 0.55, 95% CI: 0.46–0.67). The missense variant rs140185678 in *RPL3L* is also independently associated with atrial fibrillation (MAF = 0.0363, $p = 5.4 \times 10^{-9}$, OR = 1.21, 95% CI: 1.14–1.30) and atrial flutter ($p = 1.1 \times 10^{-7}$, OR = 1.20, 95% CI: 1.12–1.28). Overall, we were able to recover a small number of associations using imputed PTVs, indicating that better imputation methods are likely needed in the absence of direct genotyping of PTVs.

**Targeted PTV phenome-wide association study**. To further assess the role of PTVs across medical phenotypes, we performed a phenome-wide association analysis (pheWAS) to determine whether PTVs that have been implicated in disease predisposition may impact other diseases or commonly measured traits[58]. We focused this analysis on PTVs with minor allele frequency > 0.01% in the 17 genes with significant associations in our GWAS. In addition to PTVs in the genes identified here, we also investigated PTVs in genes with previously identified protective effects such as: *CARD9*, *RNF186*, and *IL23R* shown to confer protection against Crohn's disease and/or ulcerative colitis[1,2]; *ANGPTL4*, *PCSK9*, *LPA*, and *APOC3* shown to confer protection against coronary heart disease[4,13–17]; and *SCN9A* where homozygous PTV carriers show an inability to experience pain[59] (Supplementary Table 1).

We identified all associations ($p < 0.01$) for PTVs in these 25 genes with a MAF > 0.01% and found that PTVs in many of these genes were associated with a broad range of phenotypes (Supplementary Data 3, Supplementary Fig. 6). PTVs in eight of the 25 genes were associated with eight or more phenotypes. We observed associations between the viral receptor *IFIH1* and 10 phenotypes including protective effects against hypothyroidism, hypertension, gastric reflux, and psoriasis (Fig. 3, Supplementary Table 1). Despite minor allele frequencies ranging from 0.02% to 1.5%, three of these associations were observed for more than one *IFIH1* PTV. PTVs in *IFIH1* were also associated with increased risk for ulcerative colitis, inflammatory bowel disease, and endometriosis. We identified protective effects for *IL33* for hayfever (coded hayfever/allergic rhinitis), nasal polyps, and

angina as well as weak risk effects for bowel/intestinal obstruction and shoulder/scapula fracture (Supplementary Fig. 6). Overall, these results demonstrate that PTVs can have pleiotropic effects across diverse phenotypes and that PTVs in the same gene can both protect against and increase risk for different diseases.

We extended the pheWAS analysis to 47 sets of genes including gene sets of importance for diabetes and schizophrenia[60,61] as well as more general gene sets such as genes with associations in ClinVar and genes near GWAS peaks (Methods)[62,63]. We found several associations in important gene sets that were near significance in this study, particularly in genes near GWAS peaks (Supplementary Data 3, Supplementary Fig. 7). We also performed PTV burden tests by counting the number of PTVs present in each subject for each gene set and performing association analyses with the 135 phenotypes with more than 2000 cases. We found seven associations between gene sets and phenotypes (BY-adjusted $p < 0.05$, Supplementary Data 3). Five of the seven associations were between cancer phenotypes and gene sets that included *BRCA2* which had many PTVs on the genotyping array. These results indicate that exome sequencing may be needed to identify associations between PTV burden across multiple genes association and disease.

**Human gene knockout analysis**. Homozygous carriers of PTVs, referred to as homozygous knockouts (KOs), may have dramatically altered medical outcomes compared with carriers with only one PTV (heterozygous KOs)[64]. Genetic association analyses typically assume that genetic effects are additive; that is, the log OR of a homozygote is expected to be twice the log OR of a heterozygote. Given the large difference between having one functional copy and no functional copies of a gene, however, we expect that homozygote KOs may have non-additive effects that are stronger or weaker than would be predicted given the effect size for heterozygote KOs. To assess whether any of the 17 genes with significant associations in our GWAS or the eight genes with published protective effects (Supplementary Table 1) have evidence for non-additive effects on medical phenotypes, we estimated the KO status in each subject for each of these 25 genes. Subjects with one PTV in a gene were considered heterozygote KOs for that gene and subjects with two or more PTVs were considered homozygote KOs. In total, 16 of the 25 genes had at least one predicted homozygous KO carrier. We fit additive and non-additive models to test for associations between KO status

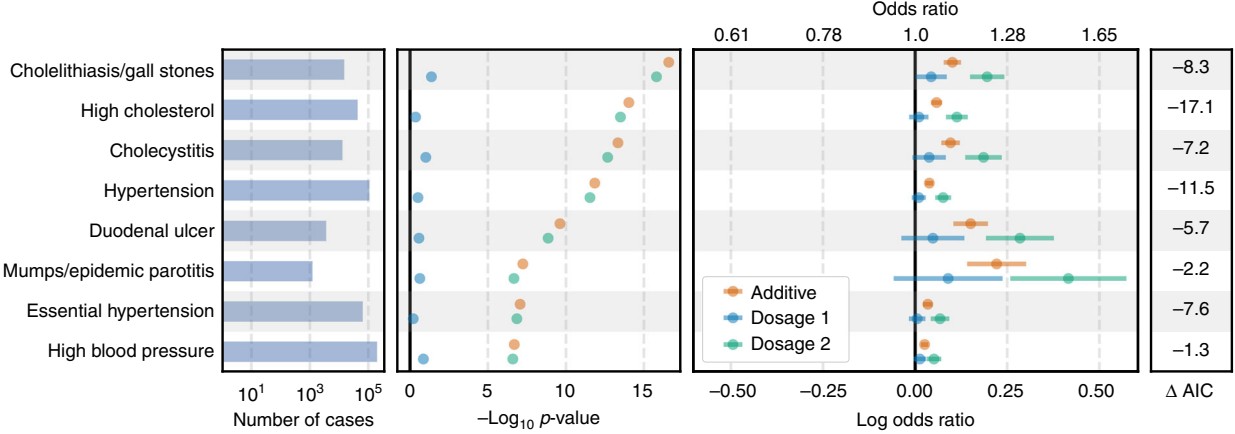

**Fig. 4** Non-additive associations for FUT2. Association results under additive and non-additive logistic regression models for predicted FUT2 heterozygous or homozygous knockouts (KOs) with a difference between non-additive model AIC and additive model AIC < −1. The left panel shows the number of cases per phenotype. The middle-left panel shows the -log₁₀ p value for the KO association analysis. The middle-right panel shows the estimated log odds ratios and 95% confidence intervals under an additive model (orange) and under a non-additive model for heterozygote KOs (blue) and homozygote KOs (green)

for these 16 and 206 medical phenotypes (minimum 1000 cases, Supplementary Fig. 8) and found 13 associations (6 distinct genes, 12 distinct phenotypes) with potential non-additive effects (Supplementary Fig. 9, Supplementary Data 5, Methods).

We identified 87,176 predicted homozygous KOs for *FUT2* caused by a common PTV rs601338 with MAF 49.1% and identified non-additive risk associations between *FUT2* KO status and eight phenotypes including hypertension and mumps (Fig. 4, Supplementary Data 5). FUT2 regulates the expression of the H antigen on the gastrointestinal mucosa and genetic variation in *FUT2* is associated with Crohn's disease[65,66], psoriasis[67], plasma vitamin B12 levels[68,69], levels of two tumor biomarkers[70,71], and urine fucose levels[72]. Under a non-additive model, the ORs for heterozygous *FUT2* KOs are all nearly one while *FUT2* homozygous KOs have ORs ranging from 1.05 (95% CI: 1.03–1.07) to 1.51 (95% CI: 1.29–1.77). Given the frequency of the rs601338 PTV, our results indicate that FUT2 function may have an important role in a wide range of phenotypes.

We also found evidence that the association between *GSDMB* KO and asthma described in our GWAS analysis above is non-additive (Figure S9, Supplementary Data 5). In total, we identified 168,025 heterozygous KOs and 74,534 homozygous KOs for *GSDMB*. Under an additive model, *GSDMB* heterozygote KOs are predicted to have a decreased risk for asthma with OR = 0.90 ($p = 5.9 \times 10^{-50}$; 95% CI: 0.88–0.91). Under a non-additive model, however, *GSDMB* heterozygote KOs are predicted to have OR = 0.86 ($p = 4.3 \times 10^{-38}$; 95% CI: 0.84–0.88) while *GSDMB* homozygote KO offers only modestly higher protection ($p = 9.7 \times 10^{-46}$, OR = 0.81, 95% CI: 0.79–0.84). Variants that increase expression of *GSDMB* in humans are associated with asthma risk,[73] and increased *GSDMB* expression causes an asthma phenotype in mice[74]. Our results suggest that knocking out just one copy of *GSDMB* provides most of the protective effect on asthma risk. Overall, we identified non-additive PTV associations for six of 16 genes tested demonstrating that the effect of PTVs on disease risk can be complex.

## Discussion

Assessing the medical relevance of protein-truncating variants is critical for prioritizing putative drug targets and clinical interpretation. We systematically characterized the association of PTVs, a class of variants with functional consequences likely to be consistent with inhibition, with medical phenotypes using data from the UK Biobank study. We estimated the effects of PTVs across 135 phenotypes and identified 27 associations between PTVs in 17 genes and 20 different phenotypes. We found four associations for PTVs with minor allele frequency < 0.1%, indicating that more subjects or case/control studies design may be needed to test for associations between ultra-rare PTVs and relatively low prevalence diseases that are not well-represented in biobank datasets. We performed 25 phenome-wide association analyses for the genes implicated by GWAS in this study plus eight genes curated from the literature (Supplementary Table 1) and identified eight genes that were associated with eight or more phenotypes (p < 0.01). Six of these 25 genes showed evidence for non-additive associations across several phenotypes including non-additive associations between *GSDMB* and asthma and *FUT2* and eight phenotypes including hypertension and cholesterol.

The genetic associations reported here directly link gene function to disease etiology and provide attractive targets for drug discovery. Naturally occurring human knockouts that protect against disease provide in vivo validation of safety and efficacy and may be relatively simple to target with drugs. Protective associations between PTVs in *IL33* and asthma; *GSDMB* and asthma; and *IFIH1* and hypothyroidism represent particularly attractive drug targets, whereas risk associations between PTVs in *FANCM* and lung cancer and *NOL3* and muscle injuries implicate these genes as important to the development of these conditions. Our results illustrate the value of deep population-scale health and genomic datasets for prioritizing genetic variants and genes with translational potential.

## Methods

**Quality control of genotype data**. We used genotype data from UK Biobank dataset release version 2 for all aspects of the study except the imputed PTV GWAS[18,75]. To minimize the impact of cofounders and unreliable observations, we used a subset of individuals that satisfied all of the following criteria: (1) self-reported white British ancestry, (2) used to compute principal components, (3) not marked as outliers for heterozygosity and missing rates, (4) do not show putative sex chromosome aneuploidy, and (5) have at most 10 putative third-degree relatives. These criteria are reported by the UK Biobank in the file "ukb_sqc_v2.txt" in the following columns respectively: (1) "in_white_British_ancestry_subset," (2) "used_in_pca_calculation," (3) "het_missing_outliers," (4) "putative_sex_chromosome_aneuploidy", and (5) "excess_relatives." We removed 151,169 individuals that did not meet these criteria. For the remaining 337,205 individuals, we used PLINK v1.90b4.4[76] to compute the following statistics for each variant: (a)

genotyping missingness rate, (b) p values of Hardy–Weinberg test, and (c) allele frequencies.

**Protein-truncating variant annotation**. We annotated 784,257 autosomal variants extracted from the mapping bim files provided by the UK Biobank using VEP version 87 and the LOFTEE plugin (https://github.com/konradjk/loftee) and identified 27,057 putative PTVs[77]. We first removed 8118 PTVs specific to the UK BiLEVE Axiom Array or with missingness > 1% among the subjects genotyped on the UK Biobank Axiom Array. Despite a missingness rate of 28% on the Axiom Biobank Array, we kept rs141992399 (*CARD9*) in the analysis. We removed 11 variants with cluster plots that indicated unreliable genotypes. We removed Affx-89018997 because the REF/ALT annotation caused problems with analysis software.

We next matched our PTVs to PTVs annotated in gnomAD (gnomad.exomes. r2.0.1.sites.vcf.gz) based on genomic position, reference, and alternate alleles and compared the allele frequencies in the UKB and gnomAD by (1) performing a Fisher's exact test using the minor allele counts from the 337,205 UKB subjects and the minor allele counts from gnomAD and (2) calculating the log odds ratio of observing the minor allele in the UKB vs. gnomAD. We stratified our PTVs by minor allele frequency into the following three bins: (0.01%, 0.1%), (0.1%, 1%), (1%, 50%). For bin (0.01%, 0.1%), we removed PTVs with Fisher $p < 1e$-7 and an absolute log odds ratio > 3. For bin (0.1%, 1%), we removed PTVs with Fisher $p < 1e$-7 and an absolute log odds ratio > 2. For bin (1%, 100%), we removed PTVs with Fisher $p < 1e$-7 and an absolute log odds ratio > 1 (Supplementary Fig. 1). In total, 123 variants were removed in this step.

There were 134 variants with MAF > 0.1% that we did match to the gnomAD exome data. We manually reviewed these variants on the gnomAD browser to determine whether they were likely to accurately type a PTV in gnomAD. In cases where the PTV was present on the gnomAD browser but was not included in the exome data, we kept the PTV in our analysis. In cases where the UKB array likely typed a non-PTV or there was no variant present on the browser, we removed the PTV from our analysis. In total, 79/134 variants were removed during in this step. 18,726 PTVs remained after filtering of which 18,228 were polymorphic. We focused on these 18,228 PTVs for subsequent analyses.

We defined the MHC region as chr6:25477797–36448354 according to the Genome Reference Consortium definition (https://www.ncbi.nlm.nih.gov/grc/human/regions/MHC?asm=GRCh37). We considered any PTV in this region or within 3,000,000 base pairs of this region (to avoid including PTVs in LD with variants in the MHC) as in or near the MHC for all analyses. We use the hg19 human genome reference throughout.

**Cancer phenotype definitions**. We combined cancer diagnoses from the UK Cancer Register with self-reported diagnoses from the UK Biobank questionnaire to define cases and controls for cancer GWAS. Individual level ICD-10 codes from the UK Cancer Register (http://biobank.ctsu.ox.ac.uk/crystal/label.cgi?id=100092), Data-Field 40006 (http://biobank.ctsu.ox.ac.uk/crystal/field.cgi?id=40006), and the National Health Service (http://biobank.ctsu.ox.ac.uk/crystal/label.cgi?id=2022), Data-Field 41202 (http://biobank.ctsu.ox.ac.uk/crystal/field.cgi?id=41202), in the UK Biobank were mapped to the self-reported cancer codes, Data-Field 20001 (http://biobank.ctsu.ox.ac.uk/crystal/field.cgi?id=20001). The mapping was performed via manual curation of ICD-10 codes for each of the self-reported cancer codes. UKB field codes for self-reported cancer were created with a tree structure such that more specific cancer subtypes (e.g., "malignant melanoma") are nested under more general categories ("skin cancer"). This tree structure was preserved in the field code to ICD-10 mapping. For example, the self-reported phenotype of "lip cancer" was mapped to its field code, 1010, and the ICD-10 codes for "malignant neoplasm of lip", C00 and C000-C009. After this mapping, individuals with an affirmative entry in one or more of the phenotype collections (self-reported cancer, cancer registry, and the NHS) were included in the case cohort for the GWAS. No secondary neoplasms were included in the cancer phenotype mappings.

**High confidence phenotype definitions**. We combined disease diagnoses from the UK National Health Service Hospital Episode Statistics with self-reported diagnoses from the UK Biobank questionnaire to define cases and controls for non-cancer phenotypes. We used the following procedure to define cases and controls for non-cancer phenotypes (referred to as "high confidence" phenotypes). ICD-10 codes (Data-Field 41202) were grouped with self-reported non-cancer illness codes (Data-Field 20002) that were closely related. This was done by first creating a computationally generated candidate list of closely related ICD-10 codes and self-reported non-cancer illness codes, then manually curating the matches. The computational mapping was performed by calculating the token set ratio between the ICD-10 code description and the self-reported illness code description using the FuzzyWuzzy python package. The high scoring ICD-10 matches for each self-reported illness were then manually curated to ensure high confidence mappings. Manual curation was required to validate the matches because fuzzy string matching may return words that are similar in spelling but not in meaning. For example, to create a hypertension cohort the code description from Data-Field 20002 ("Hypertension") was mapped to all ICD-10 code descriptions and all closely related codes were returned ("I10: Essential (primary) hypertension" and "I95:

Hypotension"). After manual curation code I10 would be kept and code I95 would be discarded.

**Family history phenotype definitions**. We used data from Category 100034 (Family history–Touchscreen–UK Biobank Assessment Centre) to define "cases" and controls for family history phenotypes. This category contains data from the touchscreen questionnaire on questions related to family size, sibling order, family medical history (of parents and siblings), and age of parents (age of death if died). We focused on Data Coding 20107: Illness of father and 20110: Illness of mother.

**Genome-wide association analyses**. We performed genome-wide logistic regression association analysis with Firth-fallback using PLINK v2.00a(17 July 2017). Firth-fallback is a hybrid algorithm which normally uses the logistic regression code described in[78], but switches to a port of logistf() (https://cran.r-project.org/web/packages/logistf/index.html) in two cases: (1) one of the cells in the $2 \times 2$ allele count by case/control status contingency table is empty (2) logistic regression was attempted as all the contingency table cells were nonzero, but it failed to converge within the usual number of steps. We used the following covariates in our analysis: age, sex, array type, and the first four principal components, where array type is a binary variable that represents whether an individual was genotyped with UK Biobank Axiom Array or UK BiLEVE Axiom Array. For variants that were specific to one array, we did not use array as a covariate. We stratified GWAS $p$ values from PLINK into three minor allele frequency bins: 0.01–0.1% (2562 PTVs), 0.1–1% (700 PTVs), and > 1% (463 PTVs). We corrected $p$ values separately for each bin using the Benjamin-Yekutieli approach implemented in R's p.adjust[79]. We considered associations with BY-corrected $p$ values < 0.05 as significant which controls the false discovery rate at 5%. As we identified 74 significant associations in our main analysis, we would expect ~ 4 false-positive associations. We also applied the Bonferroni correction for each MAF bin and for all tests for reference (Supplementary Data 3).

For the missense variant GWAS, we identified missense variants with MAF > 0.01% in each of the 17 non-MHC genes that had a significant PTV from the PTV GWAS. All genes except for *IRF5* had at least one missense variant. We then performed associations analyses as described above for the missense variants from each gene and the phenotypes that PTVs in that gene were associated with. We considered significant any missense-phenotype associations with nominal $p < 0.001$. We repeated the association analyses using the PTV genotype as a covariate to evaluate whether the association signals were independent for significant missense variants.

**HLA conditional analysis**. We performed conditional association analyses for 47 of the 74 significant associations from our GWAS for PTVs in genes in or near the MHC using the HLA alleles provided by the UK Biobank (ubk_hla_v2.txt). For each PTV-phenotype association, we re-ran the association analysis using each of the 344 HLA alleles polymorphic in the 337,205 subjects used here as a covariate in turn. We then identified which HLA allele, when used as a covariate, corresponded to the largest $p$ value for the additive genetic effect. These results are reported in Supplementary Data 4. Note that this HLA allele is not necessarily the associated with the reported trait since LD exists between different HLA alleles.

**ANKDD1B conditional analysis**. In our initial GWAS, we found associations between the PTV rs34358 in *ANKDD1B* and family history of diabetes and high cholesterol. Since *ANKDD1B* is near *HMGCR*, we performed a conditional association analysis between rs34358 and family history of diabetes and high cholesterol using the imputed genotypes for rs17238484, an intronic variant in *HMGCR* associated with cholesterol levels[37], as covariates. We found that conditioning on rs17238484 made the association between rs34358 and high cholesterol insignificant ($p = 0.052$) but that the association between rs34358 and family history was only slightly reduced from $p = 1.5 \times 10^{-7}$ to $p = 9.1 \times 10^{-5}$. We therefore decided to include this association in Supplementary Data 3.

**Conditional analysis**. We performed conditional analyses for each of the 27 PTVs outside of the MHC with significant associations. We identified all variants genotyped on the UK Biobank array within 10 kb of the PTVs that passed filtering and had MAF > 0.01%. For each variant within 10 kb of a PTV, we ran a logistic regression as described above using PLINK but added the genotype of the nearby variant as a covariate. For each PTV-phenotype association, we identified which nearby variant resulted in the largest $p$ value for the PTV association. We report this nearby variant (cond_variant), $p$ value for the PTV association (cond_p), and the MAF of the nearby variant (cond_maf) in Supplementary Data 3. For Supplementary Fig. 5, we plotted the linkage disequilibrium (LD) between the PTV and nearby variants (minimum LD 0.9) for PTVs with MAF > 1% and for which conditional analysis identified a nearby variant that reduced the $p$ value by at least one order of magnitude. For the PTV rs2884737 in *VKORC1*, we plotted variants with LD > 0.5. For rs2004640 in *IRF5*, we plotted variants with LD > 0.6. LD values were calculated using the same UK Biobank subjects used for the GWAS.

**Imputed PTVs GWAS**. We identified 962 PTVs among the UK Biobank imputed genotypes that were not multi-allelic, had MAF > 0.01%, and were not already included in our study by comparing the chromosomal coordinates and reference and alternate alleles of PTVs annotated in gnomAD to the UK Biobank positions and alleles for the UK Biobank data. We only considered PTVs in the HRC site list version 1.1 (http://www.haplotype-reference-consortium.org/site). We removed 408 imputed PTVs that had an imputation score < 0.8, missingness > 1%, or whose MAF differed substantially from the non-Finnish European MAF in gnomAD. We removed eight more imputed PTVs that were in genes near the MHC. In total we were left with 546 imputed PTVs that we stratified into the following MAF bins: 0.01–0.1% (247 PTVs), 0.1–1% (153 PTVs), and > 1% (146 PTVs). We corrected $p$ values separately for each bin using the Benjamin-Yekutieli approach implemented in R's p.adjust[79]. We assessed linkage disequilibrium between imputed PTVs and other variants using LDmatrix in LDlink[80].

For the missense variant rs140185678 (MAF = 0.0363) in *RPL3L*, we ran GWAS as described above and found that the variant was associated with associated with atrial fibrillation ($p = 5.4 \times 10^{-9}$, OR = 1.21, 95% CI: 1.14–1.30) and atrial flutter ($p = 1.1 \times 10^{-7}$, OR = 1.20, 95% CI: 1.12–1.28). We re-ran this analysis using the genotype of the *RPL3L* PTV rs140192228 as a covariate and found that the associations between rs140185678 and atrial fibrillation ($p = 4.3 \times 10^{-9}$, OR = 1.21, 95% CI: 1.14–1.29) and atrial flutter ($p = 8.4 \times 10^{-8}$, OR = 1.20, 95% CI: 1.12–1.28) were still significant. The PTV was also significant under these models for atrial fibrillation ($p = 1.1 \times 10^{-5}$, OR = 0.85, 95% CI: 0.79–0.91) and atrial flutter ($p = 2.1 \times 10^{-6}$, OR = 0.84, 95% CI: 0.78–0.90).

**Phenome-wide association analyses**. We performed pheWAS on the 17 genes with at least one significant association in our GWAS as well as 8 genes reported to have protective genetic associations: *CARD9, RNF186, IL23R, ANGPTL4, PCSK9, LPA, APOC3,* and *SCN9A* (Supplementary Table 1). We identified associations between PTVs in these genes with MAF greater than 0.01% and 135 medical phenotypes ($p < 0.01$, Supplementary Fig. 6). Four genes (*ANGPTL4, IL23R, PCSK9,* and *APOC3*) did not have any associations with $p < 0.01$ in the pheWAS.

We also report pheWAS results for gene sets from https://github.com/macarthur-lab/gene_lists and[60,61]. We plotted the $p$ values and odds ratios for associations with $p < 0.01$ between PTVs in the genes from each gene set and 135 traits with more than 2000 cases in Supplementary Fig. 7. We also performed a burden test by counting the number of PTVs present in each subject in each gene in a gene set to create a polygenic score. If a subject had more than two PTVs present in a gene, we only counted two PTVs for that gene. We regressed the polygenic score for each gene set against disease status for 135 phenotypes with > 2000 cases using logistic regression in R. We adjusted the $p$ values for each gene set using the BY method. The significant associations are reported in Supplementary Data 3. Note that owing to the rarity of PTVs, some gene sets with a small number of genes had little or no variation in the polygenic score because we observed few polymorphic PTVs in those gene sets. We have included all PTV associations with nominal $p < 0.01$ in Supplementary Data 3 ("all_phewas" tab).

**NOL3 siRNA knockdown in human skeletal muscle cells**. Adult human skeletal muscle cells (150–05 A, Sigma-Aldrich) were cultured in skeletal muscle cell growth medium (Sigma-Aldrich). For differentiation, cells were dissociated using 0.05% Trypsin-EDTA (ThermoFisher Scientific) and replated onto collagen-coated six-well plates in skeletal muscle growth medium. After 24 h, differentiation was initiated by changing medium to skeletal muscle cell differentiation medium (Sigma-Aldrich), which was subsequently exchanged every second day. We transfected differentiating skeletal muscle cells in DMEM/F-12 medium with 30 pmol siRNA against NOL3 (s301, ThermoFisher Scientific) or a scramble negative control siRNA (ThermoFisher Scientific) using the Lipofectamine RNAiMAX Transfection reagent (ThermoFisher Scientific) according to the manufacturer's protocol.

**Electrical pulse stimulation**. Four days after siRNA-treatment, we electrically stimulated skeletal muscle cells using a C-pace unit and a six-well C-Dish (IonOptix), according to the manufacturer's specifications. The protocol, which was adapted from[81], consisted of pulses of 20 ms at 10 V in a sequence of a 5 s tetanic hold through continuous pulses at 8 Hz, a 5 s delay, 5 s of pulses at 5 Hz and another 5 s delay for a total of 5 h. Unstimulated cells were exposed to the six-well C-Dish without supply of an electric current.

**RNA isolation and qRT-PCR**. Total RNA was extracted by the Trizol method. In all, 300 ng of total RNA was used to generate cDNA through the High-Capacity cDNA Reverse Transcription Kit (Applied Biosystems) in a total volume of 20 ul. Gene expression was quantified using standard TaqMan gene expression assays (NOL3 Hs01126088_g1; ACTB Hs01060665_g1; MAFbx/FBXO32 Hs01041408_m1; ThermoFisher Scientific).

**Protein analyses**. Fifteen mg of total protein was loaded onto a 4–15% poly-acrylamide gel (Bio-Rad), separated and subsequently transferred onto a PVDF membrane (Merck Millipore). The membrane was blocked in Odyssey blocking buffer (LI-COR), incubated overnight with primary antibodies; Fbx32/MAFbx

1:500 (ab168372 Lot: GR322135_2, Abcam) and GAPDH 1:1000 (sc-48167 Lot: B2112, Santa Cruz Biotechnology) as a loading control. After washing and incubation with the appropriate fluorescent secondary antibody (a-rabbit 1:5000 (925-32211 Lot: C70926_01 and a-goat 1:5000 925-32214 Lot: C50330-07), the membranes were imaged and protein quantified using the LI-COR Odyssey Fc imaging system.

We measured the activity of caspase 8 as an early apoptotic signal inhibited by NOL3. A colorimetric Caspase 8 assay kit (ab39700, Abcam) was used according to the manufacturer's protocol. To increase the efficiency of the homogenization, the homogenate was snap-frozen in liquid nitrogen prior to protein quantification and the centrifugation step performed to remove solid material was done at a lower speed. The assay quantified the cleavage of a p-nitroanilide chromophore from the sequence Ile-Glu-Thr-Asp, and the signal was measured at OD = 405 nm. The resulting values were related to the total protein content measured with the Pierce BCA protein assay kit (ThermoFisher Scientific).

**DNA fragmentation**. We analyzed the level of DNA fragmentation as a measure of the degree of apoptosis induced by the stimulation. The Cell Death Detection ELISAplus kit (Sigma-Aldrich) was used to quantify the level of cytoplasmic histone-associated DNA fragments, according to the manufacturer's specifications. In brief, the cells were lysed directly in the culture wells, scraped off the plate and centrifuged at $200 \times g$. The supernatant (cytoplasmic fraction) was loaded onto a streptavidin-coated microplate and incubated for 2 h with a biotin-labeled histone antibody and a peroxidase-conjugated DNA antibody. An ABTS substrate was subsequently added, and the enzyme-linked immunosorbent assay was read at OD = 405 nm. The level of fragmentation was related to the total amount of protein.

**Knockout status**. We estimated PTV knockout carrier status for each individual by summing the total number of PTVs present in an individual for each gene that had at least one PTV. If a PTV was predicted to effect more than one gene, we counted that PTV for each gene. If an individual was heterozygote for two different PTVs in the same gene, we considered the individual as a homozygous KO. If an individual was predicted to carry > 2 PTVs in a given gene, we set his or her count to two. We thus obtained carrier statuses for each gene in each subject that ranged from no KO, heterozygous KO, or homozygous KO. For all 18,228 predicted PTVs, we found 262 PTVs per subject on average and 1173 genes with at least one putative KO. If we restrict to only high confidence PTVs, we observe 174 PTVs per subject on average and 995 genes with at least one putative KO. If we restrict to PTVs with MAF < 1%, we observe 95 PTVs per subject on average and 778 genes with at least one putative KO.

**Additivity analyses**. To test for departures from additivity, we tested for associations between PTV carrier status and phenotype status for 16 of the 25 genes used in the pheWAS analysis that had at least one homozygote knockout and 206 phenotypes with at least 1000 cases. For each gene and phenotype, we fit two models using the glm function in R (family = "binomial"). For the additive model, we provided PTV carrier status as a numeric variable, and for the non-additive model, we provided PTV carrier status as a factor. We included age, sex, geno-typing array, and the first four principal components as covariates for both models. To identify gene-phenotype associations with suspected departures from additivity, we identified genes and phenotypes where either the additive $p$ value or homozygote KO $p$ value was < $10^{-4}$ and the difference between the non-additive model AIC and additive model AIC was < $-1$.

**URLs**. For LDlink, see https://analysistools.nci.nih.gov/LDlink/; for gnomAD browser, see http://gnomad.broadinstitute.org/; for UK Biobank, see http://www.ukbiobank.ac.uk/.

**Data availability**. The UK Biobank data are available through the UK Biobank (http://www.ukbiobank.ac.uk/). Analysis scripts and notebooks are available on Github at https://github.com/rivas-lab/public-resources. GWAS results can be browsed on the Global Biobank Engine (biobankengine.stanford.edu).

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

## Acknowledgements

This research has been conducted using the UK Biobank Resource under Application Number 24983 and 16698. We thank all the participants in the UK Biobank study. We would like to thank Stefan Stender for suggesting that the association between the PTV rs34358 and high cholesterol might be due to the LDL GWAS variant rs17238484 in *HMGCR* (https://gitter.im/UK-Biobank/Lobby). M.A.R. and C.D. are supported by Stanford University and a National Institute of Health center for Multi- and Trans-ethnic Mapping of Mendelian and Complex Diseases grant (U01 HG009080). M.A.R is supported by the GSP Coordinating Center (U24 HG008956). C.D. is supported by a postdoctoral fellowship from the Stanford Center for Computational, Evolutionary, and Human Genomics. M.A.R. is a Faculty Fellow at the Stanford Center for Population Health Sciences. Y.T. is supported by Funai Overseas Scholarship from Funai Foundation for Information Technology. M.E.L. is supported by the Knut and Alice Wallenberg Foundation. Y.T., AL, and G.M. are supported by the Stanford University Biomedical Informatics Training Program (T32 LM012409). E.A.A. is supported by 1U24EB023674-01 (MoTrPAC) and 1U01HG007708 (Undiagnosed Diseases Network). The primary and processed data used to generate the analyses presented here are available in the UK Biobank access management system (https://amsportal.ukbiobank.ac.uk/) for application 24983, "Generating effective therapeutic hypotheses from genomic and hospital linkage data" (http://www.ukbiobank.ac.uk/wp-content/uploads/2017/06/24983-Dr-Manuel-Rivas.pdf), and the results are displayed in the Global Biobank Engine (https://biobankengine.stanford.edu). We thank the Customer Solutions Team from Paradigm4 who helped us implement efficient databases for queries and application of inference methods to the data.

## Author contributions

M.A.R. conceived and designed the study. C.D., Y.T., G.M., A.L., and M.A.R. designed and carried out the statistical and computational analyses. C.C. optimized and implemented computational methods. C.D., Y.T., G.M., A.L., and M.J.D. carried out quality control of the data. Browser features in Global Biobank Engine were led and developed by G.M. and M.A.R., with assistance from A.L., Y.T., and C.D. M.E.L. and E.A.A. designed and carried out *NOL3* experiments. E.I., E.A.A., M.J.D. and C.D.B. provided analysis and commented on the manuscript. The manuscript was written by C.D., Y.T., and M.A.R. M.A.R. supervised all aspects of the study.

## Additional information

**Competing interests:** C.D.B. is a member of the scientific advisory boards for Liberty Biosecurity, Personalis, 23andMe Roots into the Future, Ancestry.com, IdentifyGenomics, and Etalon and is a founder of CDB Consulting. M.J.D. is a member the scientific advisory board for Ancestry.com. M.A.R. is a paid consultant for Genomics PLC and Prime Genomics. E.I. is a scientific advisor for Precision Wellness and Olink Proteomics for work unrelated to the present project. The remaining authors declare no competing interests.

