## [Peer Review File · Nature Communications]

Reviewer #1 (Remarks to the Author):

Summary

This was an excellent study on studying the impact of PTVs on the phenotypes that are obtainable from the UK biobank dataset and one that is very timely and relevant. After filtering and QC, they identified 18,228 PTVs that were segregating in 337,208 people, and obtained phenotypes based on the health registries and self-reported status. They then performed GWAS, PheWAS and collapsed association and additivity analysis on their candidate hits.

Phenotyping

In Figure S2, they show the numbers of cases reported for various cancers that they identified from self-reported status and those obtained from their curation of health records. This is not a criticism of the authors choice to include both sources of phenotypes, but it would be a useful addition to the paper to show the overlap between the health records and self-reported status on the same patients to understand the confidence we have in the phenotyping status.

GWAS procedure

The major comment I have on their paper is on their use of three different thresholds for MAF of the PTVs and their separate multiple hypothesis testing correction by FDR-BY. This is unusual in the field (see Sham & Purcell, Nature Reviews Genetics, 2014) at least in settings of case-control designs, and perhaps quite lenient and I would suggest that the authors report standard Bonferroni corrected statistics in their results section. I understand that there are differences in power at different MAF thresholds, as reported in Figure S3, but perhaps the authors could justify their choice of using this procedure in this setup further.

Of particular note is their use of family history information to confirm some of their results and shows that this is an effective approach at getting confidence with variants that are extremely rare.

PheWAS analysis

One of the strongest parts of the paper should have been their PheWAS analysis. As their UK biobank dataset was not initially conceived as a targeted, case control dataset for a particular phenotype, one advantage is to be able to screen across a large number of phenotypes for large effect variants (i.e. PTVs) that might have pleiotropic effects. I was a bit disappointed to see that they had restricted this analysis to a set of candidates that were first identified using their GWAS procedure. PTVs have been shown to be associated in many previous GWAS studies in a host of other diseases, and there has already been strong work in the area of neuropsychiatric-disorders by the senior authors of this study. It would be relevant here, for them to conduct their analysis on a larger gene set or for genes that are already known to affect phenotypes for which they had a large enough number of cases observed in their dataset and include all these associations as part of their results.

Homozygote carriers

Can the authors confirm if in their additivity analysis, they included compound heterozygotes, or if their assumption was genes with two or more PTVs, occurred on different haplotypes? Perhaps their results might be different if they looked at strict homozygotes in which a PTV in the same position is in homozygote stat. Further, in cases where no homozygote carriers were available, probably because these mutations were rare enough, that not one could be identified on average in a dataset of >300,000 individuals, if homozygote carriers can be observed either in the ExAC/gnomAD dataset, or the recently published studies on Pakistani subjects from consanguineous families, (Salaheen et al, Nature 2017, Narasimhan et al. Science 2016). As protective PTVs make extremely promising drug targets, outside of just additivity analysis, these would also serve as a way to check on safety of knocking out these genes in humans.

Reviewer #2 (Remarks to the Author):

I like this paper. I like the further missense variant analysis of genes with associated PTVs. I like the Phewas analysis of selected variants.

The paper is very well written and the analysis is high quality.

I have minor comments only, however I would expect these to be addressed or reasonably rebutted:

- page 1 "standing germline PTVs" - peculiar use of "standing" to this reviewer. what does it mean? suggest revise
- "predicted PTVs". Please can the accuracy of "predicted" be discussed? Perhaps especially for the splice variants.
- In the Quality Control methods, state a bit more clearly how closely related were removed.
- "hospital in-patient record data" I think it is UK NHS Hospital Episode Statistics that UK Biobank currently has. Which is not quite the same. Suggest revise.
- "identified 74 significant associations between PTVs and medical phenotype (BY-adjusted $p < 0.05$,". This might be all fine (I am not familiar with the BY method), but loses this reviewer and possibly other readers a bit. Can you add something like how many associations would have been expected to be observed by chance (maybe some phenotype label permutations can be done to assess this?).
- Add a line to explain please why the MHC excluded. I am sure this is a good thing to have done, please just explain for the reader.
- "genes in or near the MHC" please show how "near" is defined in the main text.
- the celiac risk allele is probably DQ2 (B8 in strong LD).
- I also like the Phewas of selected genes/variants. However I am unclear from the main text how multiple testing was controlled for.
- Might be worth referencing the Narasimhan Science 2016 paper which had a drug discovery analysis of human knockouts.
- some knockout papers use instead the term "loss of function". Perhaps that could be stated somewhere so readers more familiar with that term realise both PTV and "loss of function" mean the same thing. It is fine if the authors prefer to use the term PTV.
- There is an issue in UK Biobank phenotypes in that hospital data and self-reported data do not always perfectly overlap. For example myocardial infarction. I think the authors just took any report of a phenotype in either dataset and merged, but perhaps this could be explained a bit more clearly.

We would like to thank the reviewers for their thoughtful comments on our manuscript. We have made changes in the manuscript to address these points that we believe have both strengthened our results and the readability of the manuscript. Please see below for our response (in bold) to each comment (in italics). We have quoted changes in the manuscript below and bolded the changes in the manuscript as well.

Reviewer #1 (Remarks to the Author):

Summary

This was an excellent study on studying the impact of PTVs on the phenotypes that are obtainable from the UK biobank dataset and one that is very timely and relevant. After filtering and QC, they identified 18,228 PTVs that were segregating in 337,208 people, and obtained phenotypes based on the health registries and self-reported status. They then performed GWAS, PheWAS and collapsed association and additivity analysis on their candidate hits.

Phenotyping

In Figure S2, they show the numbers of cases reported for various cancers that they identified from self-reported status and those obtained from their curation of health records. This is not a criticism of the authors choice to include both sources of phenotypes, but it would be a useful addition to the paper to show the overlap between the health records and self-reported status on the same patients to understand the confidence we have in the phenotyping status.

We have added the overlap between the two phenotyping methods to Figures S2 and S6. These figures now show the number of cases obtained only from hospital records, only from questionnaire data, or from both sources.

GWAS procedure

The major comment I have on their paper is on their use of three different thresholds for MAF of the PTVs and their separate multiple hypothesis testing correction by FDR-BY. This is unusual in the field (see Sham & Purcell, Nature Reviews Genetics, 2014) at least in settings of case-control designs, and perhaps quite lenient and I would suggest that the authors report standard Bonferroni corrected statistics in their results section. I understand that there are differences in power at different MAF thresholds, as reported in Figure S3, but perhaps the authors could justify their choice of using this procedure in this setup further.

We have added Bonferroni-corrected statistics to the results in Table S3 where we also used the BY method. We have noted this in the Methods: “We also applied the Bonferroni correction for each MAF bin and for all tests for reference(Table S3).”

We chose to stratify the PTVs by MAF due to the expected power differences for variants at different MAF as shown in Figure S3. We used the BY method for false discovery rate control because it is designed to be robust to arbitrary correlation between the statistical tests. Since we are considering many phenotypes that have correlated genetic effects as well as testing multiple PTVs in some genes, we expect correlation between our

statistical tests. Given that we are (1), testing a subset of genetic variants that are likely enriched for associations, (2) looking across multiple correlated traits, and (3) testing mostly rare variants, we think that stratifying the PTVs and applying the BY method offers the best balance between controlling FDR and identifying new associations. Previous work has argued that the Bonferroni correction is likely too stringent for phenome-wide studies such as this (10.1093/bioinformatics/btq126).

Of particular note is their use of family history information to confirm some of their results and shows that this is an effective approach at getting confidence with variants that are extremely rare.

PheWAS analysis

One of the strongest parts of the paper should have been their PheWAS analysis. As their UK biobank dataset was not initially conceived as a targeted, case control dataset for a particular phenotype, one advantage is to be able to screen across a large number of phenotypes for large effect variants (i.e. PTVs) that might have pleiotropic effects. I was a bit disappointed to see that they had restricted this analysis to a set of candidates that were first identified using their GWAS procedure. PTVs have been shown to be associated in many previous GWAS studies in a host of other diseases, and there has already been strong work in the area of neuropsychiatric-disorders by the senior authors of this study. It would be relevant here, for them to conduct their analysis on a larger gene set or for genes that are already known to affect phenotypes for which they had a large enough number of cases observed in their dataset and include all these associations as part of their results.

We have included all PTV associations with $p < 0.01$ in Table S3 and added two gene set analyses to the manuscript. We have plotted pheWAS results for genes in different genes sets from the MacArthur lab gene lists (https://github.com/macarthur-lab/gene_lists), Fuchsberger et al. 2016, and Purcell et al. 2014 in Figures S5. We also performed a burden test for PTVs in the genes in each gene set versus 206 phenotypes with more than 2,000 cases. We report these results in Table S3. We describe these results in the manuscript as follows:

We extended the pheWAS analysis to 47 sets of genes including gene sets of importance for diabetes and schizophrenia (Purcell 2014, Fuchsberger 2016) as well as more general gene sets such as genes with associations in ClinVar and genes near GWAS peaks (Methods) (Landrum 2014, Welter 2014). We found a number of associations in important gene sets that were near significance in this study, particularly in genes near GWAS peaks (Table S3, Figure S5). We also performed PTV burden tests by counting the number of PTVs present in each subject for each gene set and performing association analyses with the 135 phenotypes with more than 2,000 cases. We found seven associations between gene sets and phenotypes (BY-adjusted $p < 0.05$, Table S3). Five of the seven associations were between cancer phenotypes and gene sets that included BRCA2 which had a large number of PTVs on the genotyping array. These results

indicate that exome sequencing may be needed to identify associations between PTV burden across multiple genes association and disease.

We describe the methodology in the Methods:

We also report pheWAS results for gene sets from https://github.com/macarthurlab/gene_lists and (Fuchsberger 2016, Purcell 2014). We plotted the p-values and odds ratios for associations with $p < 0.01$ between PTVs in the genes from each gene set and 135 traits with more than 2,000 cases in Figure S5. We also performed a burden test by counting the number of PTVs present in each subject in each gene in a gene set to create a polygenic score. If a subject had more than two PTVs present in a gene, we only counted two PTVs for that gene. We regressed the polygenic score for each gene set against disease status for 135 phenotypes with more than 2,000 cases using logistic regression in R. We adjusted the p-values for each gene set using the BY method. The significant associations are reported in Table S3. Note that due to the rarity of PTVs, some gene sets with a small number of genes had little or no variation in the polygenic score because we observed few polymorphic PTVs in those gene sets. We have included all PTV associations with nominal $p < 0.01$ in Table S3 ("all_phewas" tab).

Homozygote carriers

Can the authors confirm if in their additivity analysis, they included compound heterozygotes, or if their assumption was genes with two or more PTVs, occurred on different haplotypes? Perhaps their results might be different if they looked at strict homozygotes in which a PTV in the same position is in homozygote stat. Further, in cases where no homozygote carriers were available, probably because these mutations were rare enough, that not one could be identified on average in a dataset of >300,000 individuals, if homozygote carriers can be observed either in the ExAC/gnomAD dataset, or the recently published studies on Pakistani subjects from consanguineous families, (Salaheen et al, Nature 2017, Narasimhan et al. Science 2016). As protective PTVs make extremely promising drug targets, outside of just additivity analysis, these would also serve as a way to check on safety of knocking out these genes in humans.

For the additivity analysis, we did assume that an individual who was heterozygous for two different PTVs in the same gene was a homozygous KO for that gene. Since most PTVs are rare, PTVs in the same gene are not likely to be on the same haplotype. Nonetheless, exceptions do exist and we did not explicitly handle them separately. To make it clear to the readers we have added a sentence in the Methods: “If an individual was heterozygote for two different PTVs in the same gene, we considered the individual as a homozygous KO.” We observe 1,173 genes with at least one KO knockout when counting compound heterozygotes as KOs. This number drops to 1,044 when we do not count compound heterozygotes. For the genes reported in our additivity analysis, the number of homozygote KOs is mostly unaffected by not counting compound heterozygotes. We have added a column “num_ko_no_compound_hets” to Table S6 that

shows the number of homozygous KOs observed for the genes with non-additive associations when we do not include compound heterozygotes.

We have added Table S1 that compares the genes with observed homozygous loss-of-function variants in our study, Salaheen et al. 2017, Narasimhan et al. 2017, ExAC, and Icelanders from Sulem et al. 2015. In total, 3,934 genes have at least one observed homozygous loss-of-function carrier in these studies. We reference these results in the Results section: “*We observed 291 genes that had at least one observed homozygous PTV carrier in our study but had no observed homozygous loss-of-function carriers in previous studies (Table S1).*”

Reviewer #2 (Remarks to the Author):

I like this paper. I like the further missense variant analysis of genes with associated PTVs. I like the phewas analysis of selected variants.

The paper is very well written and the analysis is high quality.

I have minor comments only, however I would expect these to be addressed or reasonably rebutted:

- page 1 "standing germline PTVs" - peculiar use of "standing" to this reviewer. what does it mean? suggest revise

We have removed the word “standing” as it is not necessary. The sentence now reads: “*Although tens of thousands of germline PTVs have been identified (Rivas 2015, Lek 2016, Saleheen 2015, Narasimhan 2016, Sulem 2015), their medical relevance across a broad range of phenotypes has not been characterized.*” The point of the sentence is that many segregating PTVs have been identified in previous studies though their medical impact has not been studied broadly.

- "predicted PTVs". Please can the accuracy of "predicted" be discussed? Perhaps especially for the splice variants.

The methods for predicting PTVs are still in flux and in large part have not been systematically evaluated due to the difficulty of actually demonstrating loss-of-function *in vivo*. Narasimhan, Xue, and Tyler-Smith 2016 (10.1016/j.molmed.2016.02.006) discuss the difficulties with prediction of PTVs. We’ve added the following on the prediction of PTVs to the first paragraph of the Results section: “*While methods to predict PTVs, also referred to as loss-of-function (LoF) or knockouts variants (Narasimhan 2016, MacArthur 2010), are still being improved and validated (Narasimhan 2016a), previous work has found 70% of nonsense PTVs predicted to cause NMD have evidence for decreased expression of the corresponding transcript and 79% of splice-site variants disrupt*

splicing (Rivas 2015), indicating that predicted PTVs are likely to affect gene expression or function.”

- *In the Quality Control methods, state a bit more clearly how closely related were removed.*

We have added information to the methods that states the exact file from the UK Biobank and the specific columns that we used to filter out related individuals:

To minimize the impact of cofounders and unreliable observations, we used a subset of individuals that satisfied all of the following criteria: (1) self-reported white British ancestry, (2) used to compute principal components, (3) not marked as outliers for heterozygosity and missing rates, (4) do not show putative sex chromosome aneuploidy, and (5) have at most 10 putative third-degree relatives. These criteria are reported by the UK Biobank in the file "ukb_sqc_v2.txt" in the following columns respectively: (1) "in_white_British_ancestry_subset," (2) "used_in_pca_calculation," (3) "het_missing_outliers," (4) "putative_sex_chromosome_aneuploidy", and (5) "excess_relatives." We removed 151,169 individuals that did not meet these criteria.

- *"hospital in-patient record data" I think it is UK NHS Hospital Episode Statistics that UK Biobank currently has. Which is not quite the same. Suggest revise.*

We have updated the text to read “*hospital in-patient data (National Health Service Hospital Episode Statistics)*” which is consistent with the description from the UK Biobank (<http://biobank.ctsu.ox.ac.uk/showcase/docs/HospitalEpisodeStatistics.pdf>).

- *"identified 74 significant associations between PTVs and medical phenotype (BY-adjusted $p < 0.05$,". This might be all fine (I am not familiar with the BY method), but loses this reviewer and possibly other readers a bit. Can you add something like how many associations would have been expected to be observed by chance (maybe some phenotype label permutations can be done to assess this?).*

The BY method is a false discovery rate approach similar to the Benjamini-Hochberg (BH) procedure. Setting the BY cutoff at 0.05 means that we expect 5% of our significant results to be false positives. Since we identified 74 significant associations, we would expect ~4 false positive associations. We have noted this in the Methods: “*We considered associations with BY-corrected p -values less than 0.05 as significant which controls the false discovery rate at 5%. Since we identified 74 significant associations in our main analysis, we would expect ~4 false positive associations.*”

- *Add a line to explain please why the MHC excluded. I am sure this is a good thing to have done, please just explain for the reader.*

We've added a line in the results explaining why we focus on variants outside of the MHC: ***"Since PTVs in or near the MHC likely tag HLA risk alleles, we focused on associations for PTVs outside of the MHC."***

- *"genes in or near the MHC" please show how "near" is defined in the main text.*

We've added the MHC definition to the main text: ***"Among the 74 PTV-phenotype associations we identified, 27 involved PTVs in genes outside of the MHC (chr6-25477797-36448354)."***

We've also added information on the MHC definition to the Methods: ***"We defined the MHC region as chr6:25477797-36448354 according to the Genome Reference Consortium definition (<https://www.ncbi.nlm.nih.gov/grc/human/regions/MHC?asm=GRCh37>). We considered any PTV in this region or within 3,000,000 base pairs of this region (to avoid including PTVs in LD with variants in the MHC) as in or near the MHC for all analyses."***

- *the celiac risk allele is probably DQ2 (B8 in strong LD).*

We have updated the sentence that mentions the Celiac risk allele to read ***"However, conditioning upon the presence of HLA-B8, which is on the same haplotype as the HLA-DQ2 Celiac risk allele, reduced the p-value of the association between rs72841509 and Celiac disease to $p=0.92$ (Tjon 2010, Price 1999)."***

For the purposes of this study, we were interested in whether conditioning on any HLA allele would remove the association between a particular PTV and phenotype. Therefore, for each PTV, we report which HLA allele resulted in the largest p-value for that PTV when conditioned on the HLA allele. This demonstrates that at least one HLA allele explains the PTV association. However, this HLA allele may not actually be the HLA allele associated with the trait due to LD or other factors. We have noted this in the "HLA Conditional Analysis" section in the Methods: ***"Note that this HLA allele is not necessarily the associated with the reported trait since LD exists between different HLA alleles."***

- *I also like the Phewas of selected genes/variants. However I am unclear from the main text how multiple testing was controlled for.*

We did not explicitly control for multiple testing in the pheWAS section since we are only investigating associations for a relatively small number of PTVs. The motivation for the pheWAS section is that given a strong association between a PTV and a particular phenotype, we are interested in other phenotypes the PTV may be associated with. The initial strong association provides strong prior evidence that the PTV may be functional. Therefore, we report associations with a nominal p-value less than 0.01 for that PTV. This gives readers the opportunity to evaluate potential associations that we may not be

powered to detect here (due to number of cases for instance) in the context of other data or results.

- *Might be worth referencing the Narasimhan Science 2016 paper which had a drug discovery analysis of human knockouts.*

We have added a reference to this paper at the beginning of the results section:

“Although tens of thousands of germline PTVs have been identified (Rivas 2015, Lek 2016, Saleheen 2015, Narasimhan 2016, Sulem 2015), their medical relevance across a broad range of phenotypes has not been characterized.” We have also included the results from this paper in our comparison of genes with observed homozygous loss-of-function in Table S1.

- *some knockout papers use instead the term "loss of function". Perhaps that could be stated somewhere so readers more familiar with that term realise both PTV and "loss of function" mean the same thing. It is fine if the authors prefer to use the term PTV.*

We’ve added the following to the first paragraph of the Results section to indicate that others use the terms “loss of function” or “knockout” where we use PTV: *“While methods to predict PTVs, also referred to as loss-of-function (LoF) or knockouts variants (Narasimhan 2016, MacArthur 2010), are still being improved and validated (Narasimhan 2016a), previous work has found 70% of nonsense PTVs predicted to cause NMD have evidence for decreased expression of the corresponding transcript and 79% of splice-site variants disrupt splicing (Rivas 2015), indicating that predicted PTVs are likely to affect gene expression or function.”*

- *There is an issue in UK Biobank phenotypes in that hospital data and self-reported data do not always perfectly overlap. For example myocardial infarction. I think the authors just took any report of a phenotype in either dataset and merged, but perhaps this could be explained a bit more clearly.*

We have added the overlap between the diagnoses from hospital data and self-reported data to Figures S2 and S6. We describe how the phenotypes were defined in sections “Cancer Phenotype Definitions” and “High Confidence Phenotype Definitions” in the methods. We’ve added the following sentences at the front of those sections to clarify our overall approach: *“We combined cancer diagnoses from the UK Cancer Register with self-reported diagnoses from the UK Biobank questionnaire to define cases and controls for cancer GWAS.”* and *“We combined disease diagnoses from the UK National Health Service Hospital Episode Statistics with self-reported diagnoses from the UK Biobank questionnaire to define cases and controls for non-cancer phenotypes.”*

Reviewer #1 (Remarks to the Author):

Phenotyping

The authors have provided a clear description of the overlap of the self-reported and NHS records, GWAS procedure

I thank the authors for providing global and maf stratified bonferroni corrected p-values. I agree that there is some correlation between genes and phenotypes and the signals might be correlated but it was good to see the explicit global bonferroni corrected statistics. At this level of significance, overstrict or otherwise, it seems that we lose some of the associations. However, their approach is vindicated by their experimental follow up of the gene NOL3 which initially appears to no longer be significant at a global bonferroni p-value but as shown in the experiment has clear biological significance.

PheWAS

The authors have gone over and beyond what I and the other reviewer asked for here, performing additional burden tests for each gene set. This new analysis was clear and the methodology is sound. I am in agreement with their choice of obtaining these sets from the Macarthur lab gene lists page. I also agree with their conclusion that we need exome sequence data to improve power using this framework - something I did not appreciate in my first review.

Homozygous carriers

I thank the authors for clarifying their choice of an additional column indicating compound hets. I agree that this will make little difference on their additivity analysis. Their reporting of newly discovered homozygous PTVs is also useful.

Reviewer #2 (Remarks to the Author):

line 126 "it is critical to evaluate whether the associated variant is causal in the context of neighboring variants":

I completely agree. However I am not sure the authors have systematically checked the the causal variants are really the PTV, rather e.g. a non-coding variant close by in strong LD. This data is available for UK Biobank and some sort of conditional analysis (as the authors seem to present for some genes) should be straightforward. At least some comment needs to be made about this.

no other comments

We would like to thank the reviewers for their thoughtful comments on our manuscript. We have made changes in the manuscript to address the last point by the second reviewer. We have quoted changes in the manuscript below and bolded the changes in the manuscript as well.

Reviewer #1 (Remarks to the Author):

Phenotyping

The authors have provided a clear description of the overlap of the self-reported and NHS records,

GWAS procedure

I thank the authors for providing global and maf stratified bonferroni corrected p-values. I agree that there is some correlation between genes and phenotypes and the signals might be correlated but it was good to see the explicit global bonferroni corrected statistics. At this level of significance, overstrict or otherwise, it seems that we lose some of the associations. However, their approach is vindicated by their experimental follow up of the gene NOL3 which initially appears to no longer be significant at a global bonferroni p-value but as shown in the experiment has clear biological significance.

PheWAS

The authors have gone over and beyond what I and the other reviewer asked for here, performing additional burden tests for each gene set. This new analysis was clear and the methodology is sound. I am in agreement with their choice of obtaining these sets from the Macarthur lab gene lists page. I also agree with their conclusion that we need exome sequence data to improve power using this framework - something I did not appreciate in my first review.

Homozygous carriers

I thank the authors for clarifying their choice of an additional column indicating compound hets. I agree that this will make little difference on their additivity analysis. Their reporting of newly discovered homozygous PTVs is also useful.

We are glad that we were able to address the reviewer's comments.

Reviewer #2 (Remarks to the Author):

line 126 "it is critical to evaluate whether the associated variant is causal in the context of neighboring variants":

I completely agree. However I am not sure the authors have systematically checked the the causal variants are really the PTV, rather e.g. a non-coding variant close by in strong LD. This

data is available for UK Biobank and some sort of conditional analysis (as the authors seem to present for some genes) should be straightforward. At least some comment needs to be made about this.

no other comments

We have performed conditional analyses for the 27 significant PTV-phenotype associations for PTVs outside of the MHC. We identified all genotyped variants within 10kb of the PTV and used the genotype of those nearby variants as a covariate in the logistic regression. We found that for associations for PTVs with MAF < 1%, only one PTV had a nearby variant that could possibly account for the observed association. As expected for common variants, we found that several of the common PTVs (MAF > 1%) with associations had nearby variants that were in LD and could potentially explain the observed associations. We have added Figure S4 which shows the LD between the PTV and nearby variants. We have discussed these results in the text and cited functional studies that investigate the impact of the PTVs where appropriate:

We performed conditional analyses for the remaining 27 associations outside of the MHC by identifying genotyped variants within 10kb of the associated PTV and using the genotypes of the nearby variants as covariates for logistic regression. For PTVs with MAF less than 1%, we found that only the association between a PTV in HEATR6 and retinal detachment was explained by a nearby variant rs3744375 (Table S3). Six of the common (MAF > 1%) PTVs with associations were in high linkage disequilibrium with other nearby common variants that may explain the observed associations (Table S3, Figure S4), though the PTVs remain strong functional candidate for these associations. For instance, the gain-of-function PTV rs328 in LPL (MAF=10.1%) that we find to be associated with decreased risk for high cholesterol ($p=3.9 \times 10^{-15}$, OR=0.90, 95% CI: 0.88-0.93) and angina ($p=1.3 \times 10^{-7}$, OR=0.91, 95% CI: 0.87-0.94) and has been associated with coronary artery disease, lipid metabolism, and lower triglyceride levels (Ariza 2010, Garcia-Rios 2011, Stitzel 2016). Similarly, a recent study found that the PTV rs11078928 in GSDMB that offers protection against asthma removes exon 6 from the transcript and eliminates the ability of GSDMB to induce cell death (Panganiban 2018). The PTV rs2004640 in IRF5 has previously been associated with rheumatoid arthritis and has been connected to pathogenesis in the mouse model (Jia 2013, Weiss 2015) and the PTV rs601338 in FUT2 determines secretor status for ABH blood groups which has been associated with susceptibility to infection and several diseases (Franke 2010, Smyth 2011, Parmar 2012, Lindesmith 2003, Mottram 2017). The PTV rs2884737 in VKORC1 associated with hypertension is in moderate LD ($R^2 \approx 0.56$) with several nearby common variants and the PTV rs776746 in CYP3A5 associated with hayfever/allergic rhinitis is in near perfect LD with one other nearby variant. Additional functional work may be needed to establish whether the PTVs are causal for these two associations.

We describe the conditional analysis in the Methods section as follows:

We performed conditional analyses for each of the 17 PTVs outside of the MHC with significant associations. We identified all variants genotyped on the UK Biobank array within 10kb of the PTVs that passed filtering and had MAF > 0.01%. For each variant within 10kb of a PTV, we ran a logistic regression as described above using PLINK but added the genotype of the nearby variant as a covariate. For each PTV-phenotype association, we identified which nearby variant resulted in the largest p-value for the PTV association. We report this nearby variant (cond_variant), p-value for the PTV association (cond_p), and the MAF of the nearby variant (cond_maf) in Table S3. For Figure S4, we plotted the linkage disequilibrium (LD) between the PTV and nearby variants (minimum LD 0.9) for PTVs with MAF > 1% and for which conditional analysis identified a nearby variant that reduced the p-value by at least one order of magnitude. For the PTV rs2884737 in VKORC1, we plotted variants with LD > 0.5. For rs2004640 in IRF5, we plotted variants with LD > 0.6. LD values were calculated using the same UK Biobank subjects used for the GWAS.

Reviewer #2 (Remarks to the Author):

The authors have responded well to my comments. No further comments.

Very minor: line 550 says "17 PTVs". I think this should be 27. This could be corrected at proof stage if manuscript now accepted.

We would like to thank the reviewers for their useful comments on our manuscript. We have updated this typo in the manuscript.

Reviewer #2 (Remarks to the Author):

The authors have responded well to my comments. No further comments.

Very minor: line 550 says "17 PTVs". I think this should be 27. This could be corrected at proof stage if manuscript now accepted.

We have changed this from "17 PTVs" to "27 PTVs."